# Relationship between Health-Anxiety and Cyberchondria: Role of Metacognitive Beliefs

**DOI:** 10.3390/jcm11092590

**Published:** 2022-05-05

**Authors:** Faiza Nadeem, Najma Iqbal Malik, Mohsin Atta, Irfan Ullah, Giovanni Martinotti, Mauro Pettorruso, Federica Vellante, Massimo Di Giannantonio, Domenico De Berardis

**Affiliations:** 1Department of Psychology, University of Sargodha, Sargodha 40100, Pakistan; faiza.nadeem2018@gmail.com (F.N.); najmamalik@gmail.com (N.I.M.); mohsin.atta@uos.edu.pk (M.A.); 2Kabir Medical College, Gandhara University, Peshawar 25000, Pakistan; irfanullahecp2@gmail.com; 3Department of Neurosciences and Imaging, Chair of Psychiatry, University “G. D’Annunzio”, 66100 Chieti, Italy; giovanni.martinotti@gmail.com (G.M.); mauro.pettorruso@unich.it (M.P.); federica.vellante@gmail.com (F.V.); digiannantonio@unich.it (M.D.G.); 4National Health Service, Department of Mental Health, ASL 4, 64100 Teramo, Italy

**Keywords:** health anxiety, cyberchondria, metacognitive beliefs

## Abstract

Purpose: The current study was designed to examine the relationship between health anxiety, cyberchondria (its constructs), and metacognitive beliefs. In addition, it also evaluated the moderating role of metacognitive beliefs in this relationship. Design and Method: The present study used the purposive sampling technique to acquire a sample of (*N* = 500) adults, among them (*N* = 256) women and (*N* = 244) men, and the age of the sample ranged from 20 to 50 years. Short Health Anxiety Inventory, Cyberchondria Severity Scale, and Metacognitions Questionnaire–Health Anxiety were used to operationalize the present study variables. Findings: The descriptive statistics revealed that all instruments have good psychometric properties, as Cronbach’s alpha coefficients for all scales are ≥0.70. In addition to this, the Pearson correlation showed that all variables of the present study have a significant positive correlation with each other. Furthermore, the regression analysis described that health anxiety and metacognitive beliefs (biased thinking and beliefs about uncontrollable thoughts) were the significant positive predictors of cyberchondria. Moreover, moderation analysis showed that metacognitive beliefs significantly strengthened the association between health anxiety and cyberchondria and its constructs. Practical Implications: The present study will help medical practitioners to understand how metacognitive beliefs and health anxiety can cause an increase in cyberchondria. This will help them to design better treatment plans for people with cyberchondria.

## 1. Introduction

Being concerned about one’s health is normal. It is usual for a person to worry about his or her health when he or she is ill, for example, a person with a history of heart attacks will be more likely to feel anxious about an upcoming heart attack. In contrast to this, individuals having health anxiety tend to worry excessively about their health, and they believe that they are experiencing severe disease even in the absence of any physical symptoms. People with elevated health anxiety become distressed, adversely affecting their lives. Mostly, they do not trust their doctor’s diagnosis that they are fine and consider themselves ill [1]. Health anxiety is defined as extreme anxiety and the distress that an individual experiences because of inaccurate interpretations of his or her body sensations. Therefore, this individual believes that he or she has a severe disease. Research has demonstrated that health anxiety varies in intensity, and its extreme level is known as hypochondria [2]. Health anxiety is also referred to as illness phobia, illness anxiety, and hypochondriasis. The DSM-IV TR has a specific diagnostic criterion for health anxiety (hypochondriasis), but DSM-5 did not define it as a specific disorder, and people who were previously diagnosed with hypochondriasis are now termed as somatic symptom disorder (if they have health anxiety and somatic symptoms) and illness anxiety disorder (if they only have health anxiety in the absence of somatic symptoms) [3]. People having health anxiety are mainly worried about their body symptoms that lead to poor well-being [4]. People with an elevated level of health anxiety and chronic pain may exhibit higher tendencies to involve in safety behaviors. Safety behaviors are efforts to mitigate anxiety or avoid a feared outcome. In the case of health anxiety, this can include reassurance-seeking behaviors such as repeatedly undergoing thorough medical exams and looking up health-related information.

Moreover, such behaviors are positively associated with catastrophic thoughts about pain. In the past, people with health anxiety read books regarding different diseases to understand the disease, but in this present world, they use the Internet to get knowledge and assurance about their present health status [5]. These individuals get involved in online health-related searches and pertinently focus on the information suggesting they have a severe illness. Anxiety and assurance seeking are the main factors for using the Internet to seek health-related information [6], and that online health-related search is now known as cyberchondria. This online search seeking health-related information will not consider cyberchondria until it exceeds the normal range. Thus, the disproportionate and repetitive use of the Internet to acquire health information is known as cyberchondria [7].

The term cyberchondria first came in the U.K press in the mid-1990s, and it comprises two parts: cyber, and the second is hypochondriasis. Cyberchondria is defined as abnormal behavior in which people excessively do an online health-related search that interrupts their daily life and causes distress [8]. They are concerned about common symptomatology that may cause an increase in their anxiety. As a result, they spent more time on the internet studying health-related articles [9,10,11]. Researchers reviewed the literature (2006–2010) regarding internet usage. On behalf of this review, they said that internet users widely used the internet to get health-related information. Cyberchondria leads to exercise and healthy eating habits [12].

In contrast to this, for most people, cyberchondria causes serious health-related problems like self-diagnosis; these people also have higher chances of self-medication [13,14]. Moreover, they also share their disease-related knowledge with others, which negatively affects the lives of others as well. The present study focused on the direction of the association between cyberchondria and health anxiety. It has been found that health anxiety and cyberchondria have a bidirectional relationship. People who scored higher on health anxiety usually spent most of their time in the online health-related search. Furthermore, it has been found that this online health-related search may elevate the person’s health anxiety. Nevertheless, there is no definitive data on the direction of the relationship between cyberchondria and health anxiety [15].

The present study also evaluated the moderating role of metacognitive beliefs in the relationship between health anxiety and cyberchondria. Metacognitions are the group of strategies, information, and methods that monitor or regulate a person’s cognition [16]. The metacognitive model of psychological disorders showed that psychological disorders are linked with a particular type of thinking known as cognitive attention syndrome (CAS) [17]. The person with such thinking precisely focuses on the threats present in the internal or external environment, and metacognitive beliefs are involved in the development and maintenance of the CAS. Moreover, metacognitive beliefs have two types, i.e., positive and negative. People with positive metacognitive beliefs deliberately engage in cognitive activities that lead to CAS because they consider it beneficial [18]. Metacognitive beliefs significantly predicted emotional disorders (boredom, depression, and anxiety) and problematic internet use [19]. In addition to this, a study in Iran investigated the role of metacognitions between problematic use of the Internet and emotional dysregulation. The findings revealed that metacognitions might initiate maladaptive coping strategies, such as worry and rumination, that may cause an increase in internet usage as a source of cognitive-affective self-regulation [20].

Furthermore, metacognitions have significant positive associations with internet addiction and general health, significantly predicting health anxiety [21,22]. The present study is focused on health-related metacognitive beliefs, which have three components: biased thinking beliefs (positive metacognitive beliefs), thoughts that can cause illness, and beliefs about thought uncontrollability (negative metacognitive beliefs). Metacognitive beliefs bias a person’s attention towards positive and negative health-related information. Attention bias toward risks may make it more likely for someone suffering from health anxiety to actively seek out information regarding the ailment they fear they have [23]. Metacognitive beliefs were the significant positive predictors of health anxiety and cyberchondria [24,25,26]. People who have metacognitive beliefs (belief about biased thinking) usually believe that thinking about the illness kept them safe. Thus, they became more vigilant toward their health and acquired health-related information from the Internet. These individuals primarily focused on information that showed they had severe diseases. In addition to this, they believe that thinking positively about their health tempted their fate, and they could become ill. Such attention bias increased their health anxiety and frequency of acquiring health-related information (read online health-related articles). Moreover, the negative metacognitive beliefs (beliefs about thought uncontrollability) also moderate the relationship between health anxiety and cyberchondria. These individuals believe that they only stop worrying about their health when they get a diagnosis [25,26]. Thus, such uncontrollable thoughts lead them to online health-related searches, strengthening the relationship between health anxiety and cyberchondria. In this way, the metacognitive beliefs strengthen the association between health anxiety and cyberchondria.

### Research Gap

Based on this comprehensive theoretical background, the present study is designed to evaluate the role of metacognitive beliefs in the relationship between health anxiety and cyberchondria. Cyberchondria is an emerging mental health disorder in developing countries where most people belong to lower or middle socioeconomic status. For example, a recent study explores the cyberchondria prevalence rate in Pakistan. The study involved 985 people who came in-hospital emergencies; the results showed that about 62.17% (189) of patients used the Internet to search for their symptoms. Moreover, 11.18% of the patients did not trust physician diagnosis over their self-diagnosis.

Further, the study revealed that because of this online health-related search (cyberchondria), 80.26% of participants became panicked, and about 32 (10.52%) faced sleep problems [27]. Another study in Pakistan explored cyberchondria among healthy adults (*N* = 150) and found that about 23.3% of the participants experienced a high level of cyberchondria, while the other 26.6% reported a low level of cyberchondria [28]. In line with these, an Indian study with information technology professionals showed a high 55.6% prevalence of cyberchondria [29]. Furthermore, a study with Indian medical undergraduates showed that cyberchondria prevalence was 37.5% [30]. In low-income countries, one of the reasons to engage in this online health-related search is the expensive health care facilities and poor accessibility to hospitals. Thus, people search their symptoms online to know their treatment before visiting the physician, as all the previous work was done in high-income countries whose medical systems are far better than ours. Thus, it would be significant to understand the relationship between health anxiety, cyberchondria, and metacognitive beliefs, acknowledging that it will help medical practitioners treat cyberchondria.

Considering the abovementioned literature review, the current study hypotheses are:

**Hypothesis** **1** **(H1).***Health anxiety will be the significant positive predictor of cyberchondria*.

**Hypothesis** **2** **(H2).***Metacognitive beliefs (biased thinking and beliefs about thoughts uncontrollability) will be the significant positive predictor of cyberchondria*.

**Hypothesis** **3** **(H3).***Metacognitive beliefs will moderate the relationship between health anxiety and cyberchondria such that an increase in metacognitive beliefs will make this relationship stronger*.

(a)Metacognitive beliefs will moderate the relationship between health anxiety and compulsions (cyberchondria construct) such that an increase in metacognitive beliefs will make this relationship stronger.(b)Metacognitive beliefs will moderate the relationship between health anxiety and distress (cyberchondria construct) such that an increase in metacognitive beliefs will make this relationship stronger.(c)Metacognitive beliefs will moderate the relationship between health anxiety and excessiveness (cyberchondria construct) such that an increase in metacognitive beliefs will make this relationship stronger.(d)Metacognitive beliefs will moderate the relationship between health anxiety and reassurance (cyberchondria construct) such that metacognitive beliefs will make this relationship stronger.

## 2. Methods

### 2.1. Participants

The study took a sample of (*N* = 525) adults selected through the purposive sampling technique. After careful data screening, the researchers discarded those responses with low face validity in terms of incomplete or random responses. Finally, the principal study analysis was conducted with a sample of five hundred adults (*N* = 500), among them 51.2% women and 48.8% men. Power analysis for the sampling adequacy using A Priori method with *a* = 0.05 and a small effect size (0.02) also confirmed that the sample size of 500 is sufficient to draw inferences from data in the present study. The age of the sample ranged from 20–50 years. Moreover, data was collected from educated adults (8.4% intermediate, 78.6% graduate, and 13% postgraduate), and no participant was below the intermediate level. Further, 64.4% of participants belonged to the nuclear family system, and 35.6% belonged to the joint family system. In addition to this, about 64.6% of participants belonged to urban areas, while 35.4% belonged to rural areas.

### 2.2. Measures

#### 2.2.1. Assessment of Health Anxiety

Short Health Anxiety Inventory has been used to measure health anxiety. The Short Health Anxiety Inventory has 18-item. It has a 4-point Likert-type rating scale that ranges from 0 to 3. Among them, each item has its own response choice. A study recommended using the first 14 items of SHAI to measure health anxiety [31]. Another study [26] also used the first 14 items to access health anxiety in line with these. The present study will also use the SHAI first 14 items. The scale has demonstrated good levels of reliability, as reported by the author (α = 0.93) [32].

#### 2.2.2. Assessment of Cyberchondria

The Cyberchondria Severity Scale is a 33-item scale that has five subscales. These subscales measure the compulsion, distress, excessiveness, reassurance-seeking, and mistrust of medical professionals. The response format of the scale contained a 5-point Likert-type scale (1 = *never* to 5 = *always*). Different studies have indicated that the mistrust scale did not assess the same construct as the other CSS subscales [33]. In addition to this, another study recommends excluding the mistrust subscale (items 9, 28, 33). Due to such recommendations, the three mistrust items are not included in the present study. The compulsion subscale contains eight items; the distress subscale contains eight items; the excessiveness subscale contains eight items, and the reassurance subscale comprises six items. The Cronbach alpha computed for this scale by the author was 0.94. Moreover, the reliability of the compulsion subscale was 0.95, the distress subscale was 0.92, the excessive subscale was 0.85, and the reassurance subscale was 0.89, respectively [9].

#### 2.2.3. Assessment of Metacognitive Beliefs

Metacognitions Questionnaire-Health Anxiety is a 14-item scale. It accesses the three metacognitive beliefs related to health anxiety. One of them is known as “beliefs about thoughts can cause illness” (MCQ-C: 5 items); the second is known as “beliefs about biased thinking” (MCQ-BT: 5 items), and the third is known as “beliefs about thoughts are uncontrollable” (MCQ-U:4 items). Each item is rated on the 4-point Likert type scale 1 = *Do not agree* and 4 = *Agree very much.* The scale has good internal consistency as reported by author (α = 0.89) biased thinking (α = 0.83); beliefs about thoughts are uncontrollable (α = 0.81) and beliefs about thoughts can cause illness (α = 0.78) [22].

### 2.3. Procedure

The present study was executed following American Psychological Association’s ethical guidelines. Formal approval to conduct the study was obtained from the institutional review board of the University of Sargodha vide letter no. SU/PSY/763-b, dated: 29 July 2020. The authors’ formal permission to use scales to measure relevant constructs was also ensured. After their approval, the participants of the present study were contacted personally. As the present study focused on online health-related searches, participants were asked about their searching habits. After ensuring that they met the criteria for inclusion, they were directed to the current study objectives. They were also informed about the selection process and requested participation. Moreover, they were ensured about the confidentiality and privacy of their responses. Participants were also briefed about the study objective, and they could ask questions in case of any ambiguity. Now the scales were given to the consented participants in a booklet form. The booklet contained the informed consent, demographic sheet, and the study’s main instruments. Finally, all participants were warmly thanked for their time, sharing of information, and patience. Further, the collected data were carefully reviewed for data cleaning purposes. Incomplete or random responded questionnaires were discarded.

### 2.4. Statistical Analysis

All demographic and clinical variables in the present study were checked for deviation from the Gaussian distribution using the Kolmogorov–Smirnov test. Descriptive statistics (means and standard deviations as appropriate) were calculated on demographic variables and all psychometric scales for the study sample. Scale reliability was calculated using Cronbach’s alpha. SPSS 22-Version was used to analyze the data. Pearson’s correlation coefficient was calculated to find out the correlation between variables. Linear and multiple regression analyses were conducted to test the hypotheses to determine causal relationship pattern, prediction, and moderation analysis through PROCESS 3.5 by Hayes. A two-tailed *p* value < 0.05 was considered statistically significant for all tests.

## 3. Results

Table 1 describes all variables’ means, standard deviation, and alpha reliability. The descriptive statistics reveal that all instruments have good psychometric properties, as Cronbach’s alpha coefficients for all scales are ≥0.70. Pearson’s correlation shows that cyberchondria has a significant positive correlation with health anxiety and metacognitive belief. Further, it also demonstrates the significant positive correlation between health anxiety and metacognitive beliefs.

Results of linear regression analysis in Table 2 demonstrate that health anxiety (*β* = 0.63, *p* < 0.001) is the significant positive predictor of cyberchondria. Moreover, the analysis shows that the overall model is significant {*F* (1, 498) = 327.73, *p* < 0.001}, and it indicates that 40% of variance in cyberchondria is contributed by health anxiety (*R*^2^ = 0.40).

Table 3 [1] illustrations that metacognitive beliefs (*β* = 0.72, *p* < 0.001) are the significant positive predictor of cyberchondria. Moreover, the overall model is found to be significant {*F* (1, 498) = 547.16, *p* < 0.001} and it reveals that 52% variance in cyberchondria has occurred because of metacognitive beliefs (*R*^2^ = 0.52). Table 3 [2] demonstrates that metacognitive beliefs constructs, i.e., biased thinking (*β* = 0.40, *p* < 0.001) and thoughts are uncontrollable (*β* = 0.31, *p* < 0.001) positively predicts cyberchondria, whereas thoughts can cause illness (*β* = 0.08, *p* > 0.05) does not predict cyberchondria. Moreover, the overall model found to be significant {*F* (3, 496) = 193.93, *p* < 0.001}, and it discloses that 54% variance in cyberchondria occurs because of biased thinking and thoughts are uncontrollable (*R*^2^ = 0.54).

Table 4 describes the moderating role of metacognitive beliefs between health anxiety and cyberchondria. Findings show that metacognitive beliefs significantly moderate the relationship between health anxiety and cyberchondria in adults {*F* (3, 496) = 264.58, *p* < 0.001, *t* = 3.81}. The value of *R*^2^ = 0.62 showed that predictors explain 62% variance in cyberchondria. The significant interaction effect suggested that a positive association between health-related anxiety and cyberchondria became stronger when metacognitive beliefs increased. The interaction is schematically presented in Figure 1.

Figure 1 showed the graphic presentation of moderation analysis where steep lines indicated that the high level of metacognition beliefs strengthened the positive relationship between health anxiety and cyberchondria.



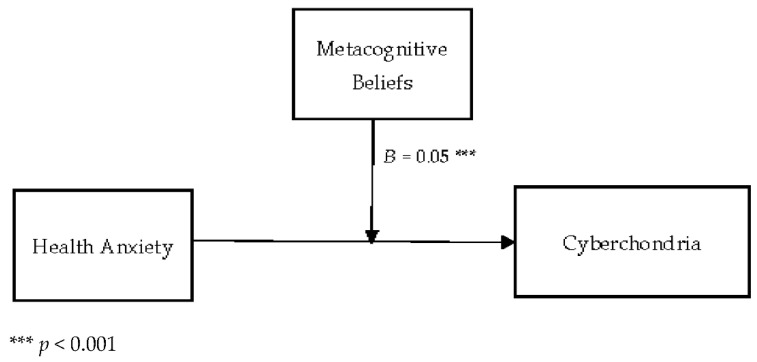



Conceptual diagram of a simple moderation model in which metacognitive beliefs have influenced the effect of health anxiety on the cyberchondria.

Table 5 describes the moderating role of metacognitive beliefs between health anxiety and compulsions (cyberchondria). Findings showed that metacognitive beliefs significantly moderate the relationship between health anxiety and compulsions in adults {*F* (3, 496) = 220.11, *p* < 0.001, *t* = 3.01}. The value of *R*^2^ = 0.57 showed that predictors explain a 57% variance in compulsions. The significant interaction effect suggested a positive association between health-related anxiety and compulsions became stronger when metacognitive beliefs increased. The interaction is schematically presented in Figure 2.

Figure 2 showed the graphic presentation of moderation analysis where steep lines indicated that the high level of metacognition beliefs strengthened the positive relationship between health anxiety and compulsions.



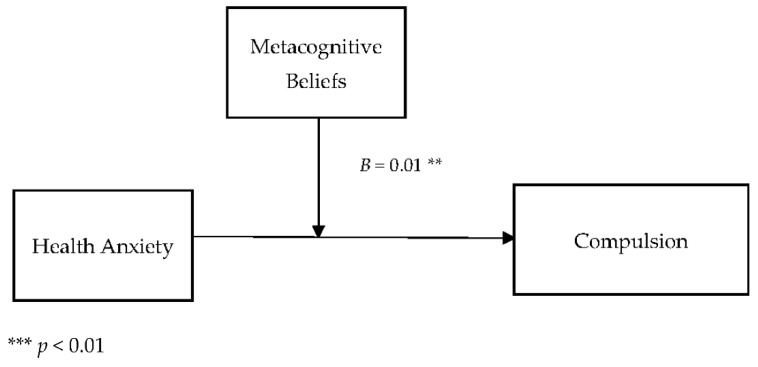



Conceptual diagram of a simple moderation model in which metacognitive beliefs have influenced the effect of health anxiety on compulsion.

Table 6 describes the moderating role of metacognitive beliefs between health anxiety and distress (cyberchondria). Findings showed that metacognitive beliefs significantly moderate the relationship between health anxiety and distress in adults {*F* (3, 496) = 234.45, *p* < 0.001, *t* = 3.63}. The value of *R*^2^ = 0.59 showed that predictors explain a 59% variance in distress. The significant interaction effect suggested that the positive association between health-related anxiety and distress became stronger when metacognitive beliefs increased. The interaction is schematically presented in Figure 3.

Figure 3 showed the graphic presentation of moderation analysis where steep lines indicated that the high level of metacognition beliefs strengthened the positive relationship between health anxiety and distress.



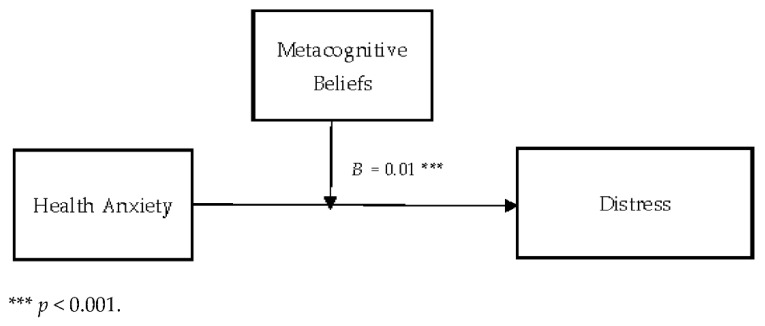



Conceptual diagram of a simple moderation model in which metacognitive beliefs have influenced the effect of health anxiety on distress.

Table 7 describes the moderating role of metacognitive beliefs between health anxiety and excessiveness (cyberchondria). Findings showed that metacognitive beliefs significantly moderate the relationship between health anxiety and excessiveness in adults {*F* (3, 496) = 228.68, *p* < 0.001, *t* = 3.56}. The value of *R*^2^ = 0.58 showed that predictors explain 58% variance in excessiveness. The significant interaction effect suggested a positive association between health-related anxiety and excessiveness became stronger when metacognitive beliefs increased. The interaction is schematically presented in Figure 4.

Figure 4 showed the graphic presentation of moderation analysis where steep lines indicated that the high level of metacognition beliefs strengthened the positive relationship between health anxiety and excessiveness.



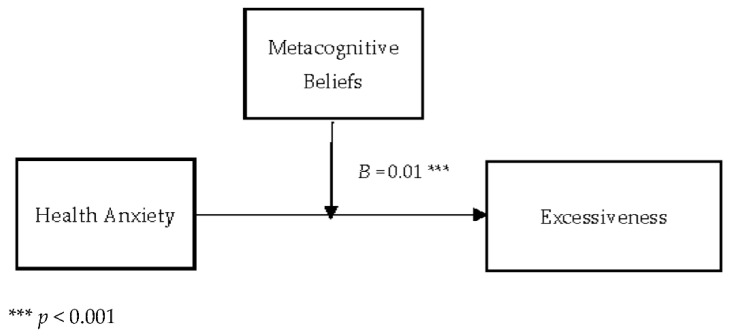



Conceptual diagram of a simple moderation model in which metacognitive beliefs have influenced the effect of health anxiety on excessiveness.

Table 8 describes the moderating role of metacognitive beliefs between health anxiety and reassurance (cyberchondria). Findings showed that metacognitive beliefs significantly moderate the relationship between health anxiety and reassurance in adults {*F* (3, 496) = 249.87, *p* < 0.001, *t* = 4.29}. The value of *R*^2^ = 0.60 showed that predictors explain 60% variance reassurance. The significant interaction effect suggested a positive association between health-related anxiety and reassurance became stronger when metacognitive beliefs increased. The interaction is schematically presented in Figure 5.

Figure 5 showed the graphic presentation of moderation analysis where steep lines indicated that the high level of metacognition beliefs strengthened the positive relationship between health anxiety and reassurance.



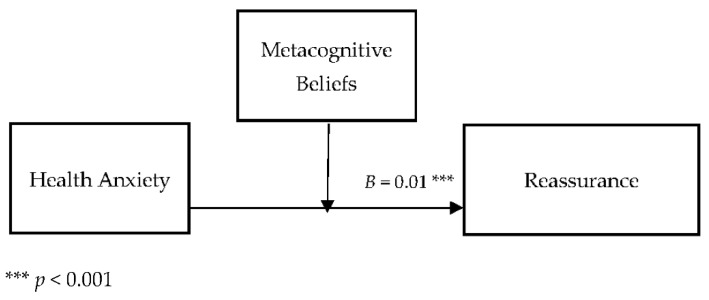



Conceptual diagram of a simple moderation model in which metacognitive beliefs have influenced the effect of health anxiety on reassurance.

## 4. Discussion

The present study evaluated the relationship between cyberchondria, health anxiety, and metacognitive beliefs. The results revealed that health anxiety has a significant positive association with cyberchondria and metacognitive beliefs (see Table 1). Keeping in view the result of correlation analysis, regression analysis was performed. Analysis revealed that health anxiety is a significant positive predictor of cyberchondria (see Table 2), supporting the first hypothesis. Furthermore, it revealed that people who have a high level of health anxiety easily misinterpret their body sensations and belief that they have a severe illness. Therefore, to become sure about their health and overcome their anxiety, they get information from different sources. In the present era, the Internet is the prime source of information, so individuals with a high level of health anxiety will engage in internet surfing to know the possible causes of these symptoms [6,34,35]. Unfortunately, such online health-related searches sometimes lead to negative outcomes (i.e., increased distress, worry) and further elevate the person’s level of health anxiety, which increases the frequency of online health-related information seeking [36,37,38,39]. Moreover, a recent study in Poland evaluated the vulnerability factors for cyberchondria among adults. The study revealed that health anxiety and low self-esteem were the most critical factors leading a person to excessive online health-related search [40,41]. Thus, the person reads different online health-related articles, blogs, and books to assure his health and reduce health anxiety.

Metacognitive beliefs were a significant risk factor for addictive behavior, such as addiction to alcohol and problematic use of the Internet [42]. Analysis depicted that cyberchondria was positively predicted by metacognitive beliefs (see Table 3), which supported the second hypothesis of the current study. The findings suggested that metacognitive beliefs such as biased thinking and beliefs about thoughts’ uncontrollability (metacognitive constructs) significantly predicted the cyberchondria (see Table 3). People with non-productive thinking styles (biased thinking and belief about thought uncontrollability) precisely focused on the threat, and such thinking styles played a significant role in the etiology of health anxiety, depression, and other negative emotions [43,44]. In line with these, it has been found that metacognitive beliefs about the thought’s uncontrollability have a unique relationship with cyberchondria. Such people believed that they did not stop thinking about their illness, which led them toward cyberchondria. Thus, such beliefs trigger the maladaptive cycle of self-regulation that causes negative emotions (worry, sadness, and anxiety) and increased problematic internet use [45,46]. In addition, metacognitive beliefs such as biased thinking are more common in the general population. It has been found that judgment biases (biased thinking) significantly predicted cyberchondria [47,48]. A recent study in Italy showed that adults’ metacognitive beliefs (thought uncontrollability and biased thinking) are the most important predictors of cyberchondria [49].

The findings also provided empirical support for the third hypothesis (see Table 4; Figure 1), as it was hypothesized that metacognitive beliefs strengthened the relationship between health anxiety and cyberchondria. The results revealed that the relationship between health anxiety and cyberchondria was stronger when metacognitive beliefs were considered. Metacognitive beliefs such as biased thinking and beliefs about thought uncontrollability played a central role in developing and maintaining health anxiety [50]. Health anxiety has been an essential predictor of cyberchondria, and it has a strong association with online health-related searches. While the metacognitive beliefs played a significant role in the maintenance and the escalation of health anxiety and such escalation of health anxiety increased the frequency of online health-related searches. In addition to this, a study showed that metacognitive beliefs have a significant positive correlation with health anxiety and catastrophic misinterpretation. Furthermore, the result revealed that the association between health anxiety and catastrophic misinterpretation has become more robust when metacognitive beliefs are added to the model [46]. These results also supported the present findings and showed that metacognitive beliefs strengthen the relationship between health anxiety and cyberchondria.

The current study also explored the role of metacognitive beliefs between health anxiety and cyberchondria constructs. The findings also provided the empirical support for the 3rd (a) hypothesis (see Table 5 and Figure 2), as it was hypothesized that metacognitive beliefs strengthen the relationship between health anxiety and compulsions (cyberchondria constructs). The results revealed that the relationship between health anxiety and compulsions held more strongly when the metacognitive beliefs were considered. These findings are consistent with previous research, as metacognitive beliefs are significantly associated with the online health-related search. In addition to this, it has been found that metacognitive beliefs increased the frequency of online health-related searches [26], and it negatively affected the person’s daily life and online activities. Thus, individuals may spend most of their time in online health-related searches, adversely affecting their studies, relationships, and jobs [51].

In addition to this, the present study also explored the moderating role of metacognitive beliefs in the relationship between health anxiety and distress (cyberchondria constructs). The present study showed that the positive relationship between health anxiety and distress became stronger when the metacognitive beliefs increased (see Table 6 and Figure 4). These findings supported the 3rd (b) hypothesis of the present study. People with health anxiety are apprehensive about their health, and they believe that they have serious health issues. This kind of worry increased their online health-related search [52]. Moreover, the study showed that such online health-related searches mostly have negative outcomes (i.e., increased distress and worry) and further elevate the person’s health anxiety level. In line with these findings, a study showed that metacognitive beliefs moderated the relationship between anxiety and stress as a high level of metacognitive beliefs strengthens the relationship between stress and anxiety [53]. Moreover, it is found that metacognitive beliefs increase a person’s anxiety sensitivity among health anxious people. Because of it, this person may consider the regular physical changes as threatening, which can cause an escalation in his or her anxiety [24].

Further, the present study also explored the moderating role of metacognitive beliefs in the relationship between health anxiety and excessiveness (cyberchondria construct). The results showed a positive relationship between health anxiety and excessiveness held strongly when the metacognitive beliefs increased (see Table 7 and Figure 4). Results supported the 3rd (c) hypothesis of the present study. Finally, results supported the 3rd (d) hypothesis that metacognitive beliefs would significantly moderate the relationship between health anxiety and reassurance (cyberchondria construct). The findings revealed that the positive relationship between health anxiety and reassurance became stronger (see Table 8 and Figure 5). The literature also supported both findings as anxious health people (people who have elevated levels of health anxiety) mostly visited health-related websites to get assurance about their health [54].

Moreover, they mainly focused on the catastrophic outcome, which further increased their health anxiety rather than decreasing it. A study showed that such escalation of health anxiety was significantly associated with excessiveness (cyberchondria construct), increasing the frequency of online health-related searches [39]. In addition to this, this escalation of health anxiety also increased the tendency to acquire expert assistance, and a person visited his or her doctor more frequently [9]. The empirical search showed that metacognitive beliefs (biased thinking, thought uncontrollability, and thoughts that cause illness) were significantly involved in developing psychological disorders [47]. Moreover, they played a significant role in developing and maintaining health anxiety, so an increased level of metacognitive beliefs strengthens the association between health anxiety and cyberchondria constructs (excessiveness and reassurance). The empirical search showed that metacognitive beliefs strengthen the association between health anxiety and cyberchondria in multiple ways. An individual with positive metacognitive beliefs or beliefs about biased thinking tends to believe that he/she may be safe if thinking about illness. Thus, these individuals are more health-conscious and acquire pertinent information. Moreover, the person who believes that thinking positively about his or her health lures his or her fate, and he or she will become ill may purposely focus on the devastating information and illness-related symptoms. This little attention might increase his or her health anxiety besides the frequency of gaining health-related information, i.e., reading online health-related articles and other material. At the same time, the negative metacognitive beliefs (beliefs about thought uncontrollability) also influence the relationship between health anxiety and cyberchondria. Thus, the individual believes that an online search of disease-related information will be helpful for him or her to stop worrying about their health while having a diagnosis. Thus, the person intends to control his or her thoughts by seeking reassurance from online health-related articles. However, these online health-related articles will further increase the person’s health anxiety [49,55].

### 4.1. Implications

The present study revealed how an emerging problem, cyberchondria, may be related to health anxiety and metacognitive beliefs. The current study is helpful for psychologists in a comprehensive understanding of cyberchondria and its constructs. This study is also beneficial for medical practitioners as it shows that people who indulge in an online health-related search might usually mistrust their physicians and question their capabilities. As a result, they consider the online information more trustworthy, which may increase their tendencies toward self-medication that further worsen the person’s condition. Moreover, self-diagnosis might increase sleep problems, and such people will become more panicked. Therefore, physicians must inform their patients about the negative effects of self-medication and the unauthenticated information available on different health-related websites during treatment. Moreover, it also highlighted the need for the government to take measures for the accountabilities of these medical websites, and only those websites should be allowed to publish health-related articles that licensed practitioners supervise.

### 4.2. Limitations

Regardless of the usefulness of the present study, it has certain limitations that future researchers should consider. Firstly, purposive sampling may lead to bias when generalizing findings, so a random sampling technique is recommended to explore the phenomenon further. Secondly, the present study used self-reported measures to investigate the present study variables. It has been found that self-reported measures have a higher chance of social desirability. People mostly try to present their positive and soft image [56,57]. Moreover, the sample was confined to those only with minimum intermediate qualifications, rendering many other potential individuals ineligible for the present study. Future researchers are suggested to consider the adults irrespective of their education. It is also suggested that future researchers use multimethod approaches that may be a combination of a structured interview and self-report measures. The researcher should interview the participants to get a detailed picture of their online health-related information-seeking behavior. Moreover, interviews with the family members of the participants may help cross-check the information.

The current study used cross-sectional survey research designed to collect the data. However, as we know, this research design did not answer all queries. Therefore, the researchers should conduct a longitudinal study rather than the cross-sectional one in the future.

## 5. Conclusions

The present study has been designed to investigate the relationship between health anxiety and cyberchondria besides the role of metacognitive beliefs. The study showed that all instruments used in the present study were applicable in Pakistani culture and had good alpha reliabilities. The correlational analysis depicted that health anxiety, cyberchondria, and metacognitive beliefs have a significant positive association. Furthermore, the prediction analysis showed that health anxiety and metacognitive beliefs were the significant positive predictors of cyberchondria. In addition to this, moderation analysis revealed that metacognitive beliefs strengthen the positive relationship between health anxiety and cyberchondria (and its constructs).

The present study might be helpful for physicians and psychologists in dealing with individuals experiencing cyberchondria by developing an insight that their online health-related search only feeds up their anxiety. Further, it has been found that health anxiety significantly predicts cyberchondria, so the psychologist should take steps (cognitive behavioral therapy) to reduce health anxiety. Moreover, the present study showed that metacognitive beliefs played a significant role in developing cyberchondria. Thus, the present study is helpful for the clinical psychologists to get a better understanding of cyberchondria.

## Figures and Tables

**Figure 1 jcm-11-02590-f001:**
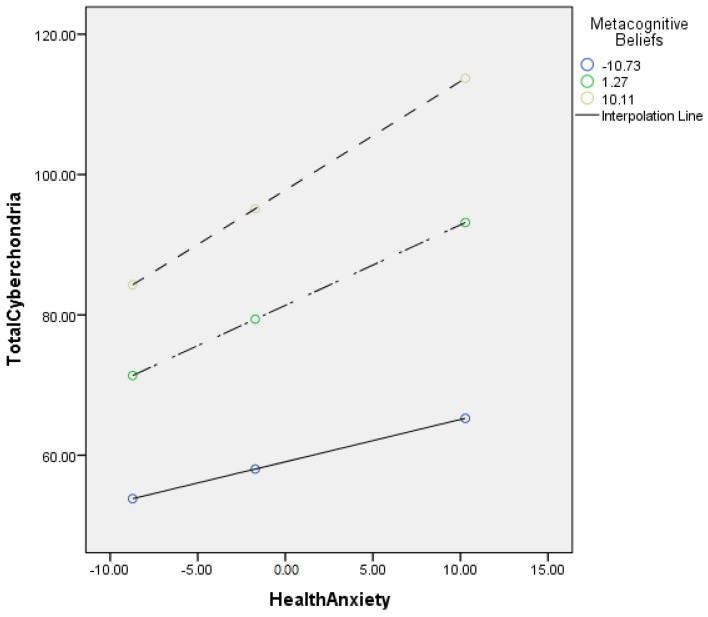
Graphical presentation of moderating role of metacognitive beliefs between health anxiety and cyberchondria.

**Figure 2 jcm-11-02590-f002:**
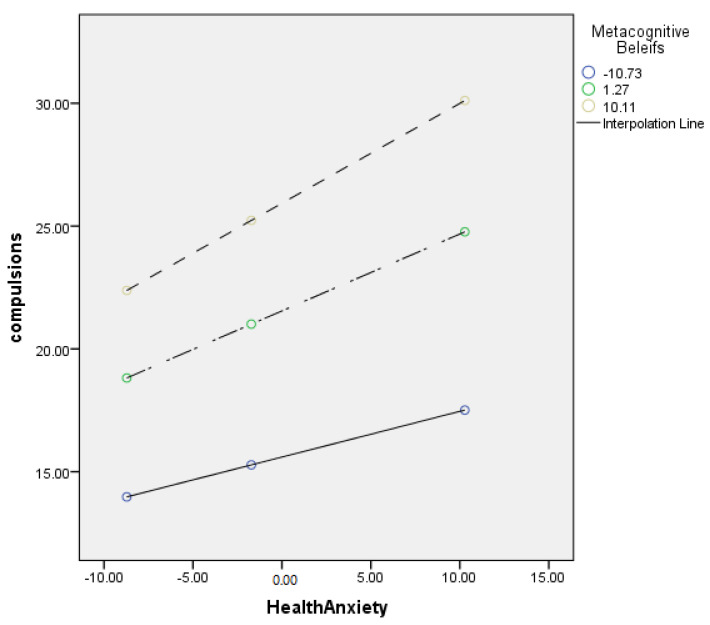
Graphical presentation of moderating role of metacognitive beliefs between anxiety and compulsions.

**Figure 3 jcm-11-02590-f003:**
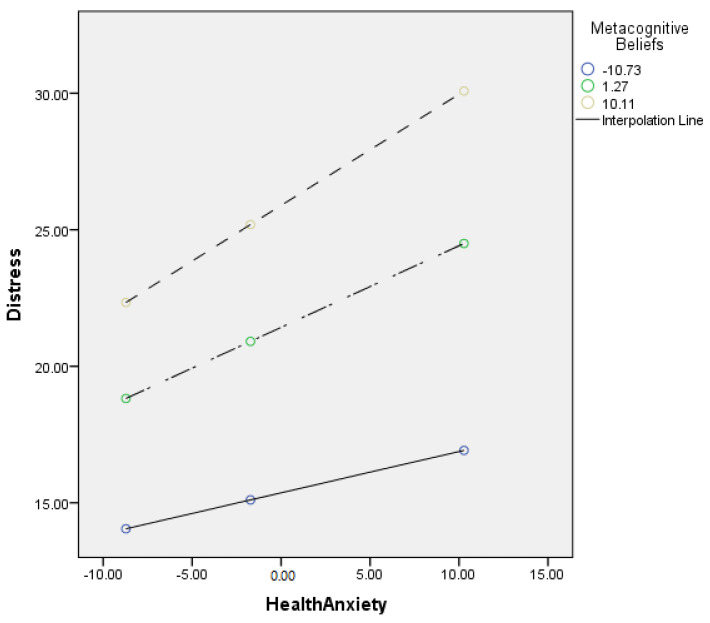
Graphical presentation of the moderating role of metacognitive beliefs between health anxiety and distress.

**Figure 4 jcm-11-02590-f004:**
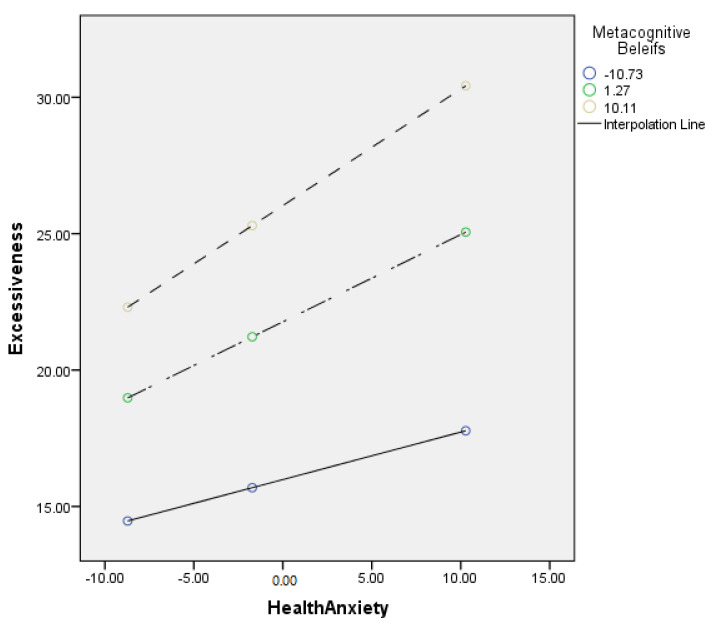
Graphical presentation of moderating role of metacognitive beliefs between health anxiety and excessiveness.

**Figure 5 jcm-11-02590-f005:**
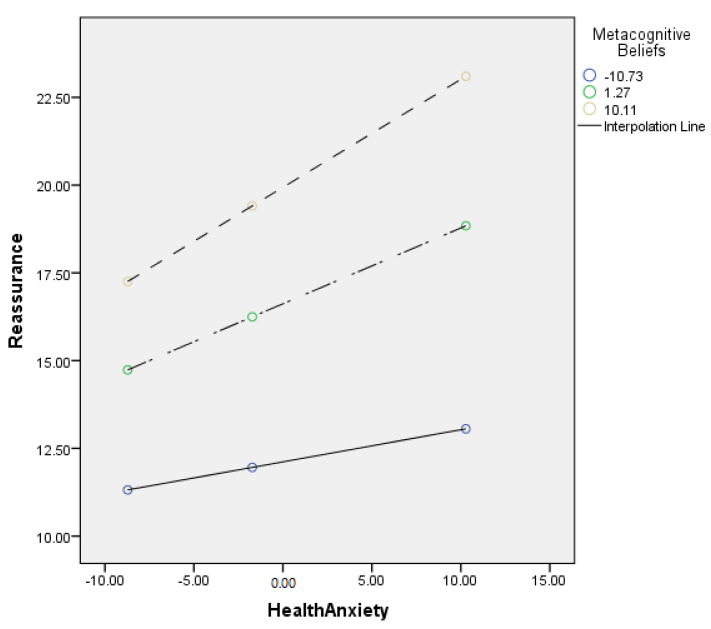
Graphical presentation of moderating role of metacognitive beliefs between health anxiety and reassurance.

**Table 1 jcm-11-02590-t001:** Means, standard deviations and correlations among health anxiety, cyberchondria (its constructs), (biased thinking, beliefs about thought uncontrollability, beliefs about thought can cause illness) and metacognitive beliefs (*N* = 500).

Variables	M	SD	α	1	2	3	4	5	6	7	8	9	10
HA	18.72	8.77	0.91		0.72 ***	0.73 ***	0.70 ***	0.72 ***	0.70 ***	0.67 ***	0.66 ***	0.66 ***	0.68 ***
CSS	80.93	29.84	0.98			0.99 ***	0.99 ***	0.99 ***	0.98 ***	0.76 ***	0.76 ***	0.75 ***	0.77 ***
CMP	21.37	8.33	0.94				0.97 ***	0.98 ***	0.94 ***	0.74 ***	0.74 ***	0.73 ***	0.76 ***
DIS	21.31	8.19	0.94					0.97 ***	0.95 ***	0.75 ***	0.75 ***	0.73 ***	0.77 ***
EXC	21.67	8.17	0.94						0.94 ***	0.75 ***	0.75 ***	0.74 ***	0.77 ***
REA	16.58	5.95	0.90							0.76 ***	0.76 ***	0.75 ***	0.78 ***
BIA	12.58	3.86	0.83								0.94 ***	0.93 ***	0.98 ***
TUA	8.64	2.86	0.75									0.90 ***	0.97 ***
TCI	11.51	3.25	0.71										0.97 ***
MCB	32.73	8.99	0.90										

Note. HA = health anxiety; CSS = cyberchondria (CMP = compulsion; DIS = distress; EXC = excessiveness; REA = reassurance); MCB = metacognitive beliefs (BIA = biased thinking; TUA = thoughts are uncontrollable; TCI = thoughts can cause illness). *** *p* < 0.001.

**Table 2 jcm-11-02590-t002:** Regression analysis for predicting cyberchondria from health anxiety (*N* = 500).

Predictor Variables	*Β*	*R* ^2^	*F*
Health Anxiety	2.14 ***	0.40	327.73 ***

*** *p <* 0.001.

**Table 3 jcm-11-02590-t003:** Regression analysis of metacognitive beliefs and its constructs (biased thinking; thoughts are uncontrollable and thoughts can cause illness) as predictors of cyberchondria (*N* = 500).

Sr. No.	Predictor Variables	*B*	*R* ^2^	*F*
(1)	Metacognitive Beliefs	2.40 ***	0.52	547.16 ***
(2)	Biased Thinking	3.13 ***		
	Thoughts are Uncontrollable	0.31 ***	0.54	193.93 ***
	Thoughts can Cause Illness	0.72		

Note. (1) = linear regression; (2) = multiple regression, *** *p* < 0.001.

**Table 4 jcm-11-02590-t004:** Moderation of metacognitive beliefs between health anxiety and cyberchondria (*N* = 500).

Predictors	*Β*		Outcome: Cyberchondria	
			95% CI	
			LL	UL
(constants)	79.00 ***		77.091	80.91
Health Anxiety	1.09 ***		0.87	1.31
Metacognitive Beliefs	1.86 ***		1.64	2.07
Health Anxiety × Metacognitive Beliefs	0.05 ***		0.02	0.07
*R* ^2^	0.62			
*F*		264.58 ***		

*** *p* < 0.001. Abbreviations: CI, confidence interval; LL, lower limit; UL, upper limit.

**Table 5 jcm-11-02590-t005:** Moderation of metacognitive beliefs between health anxiety and compulsions (cyberchondria construct) (*N* = 500).

Predictors	*Β*		Outcome: Compulsions	
			95% CI	
			LL	UL
(constants)	20.92 ***		20.35	21.48
Health Anxiety	0.30 ***		0.23	0.37
Metacognitive Beliefs	0.49 ***		0.43	0.56
Health Anxiety × Metacognitive Beliefs	0.01 **		0.004	0.02
*R* ^2^		0.57		
*F*		220.11 ***		

** *p* < 0.01. *** *p* < 0.001. Abbreviations: CI, confidence interval; LL, lower limit; UL, upper limit.

**Table 6 jcm-11-02590-t006:** Moderation of metacognitive beliefs between health anxiety and distress (cyberchondria construct) (*N* = 500).

Predictors	*Β*		Outcome: Distress	
			95% CI	
			LL	UL
(constants)	20.78 ***		20.24	21.33
Health Anxiety	0.28 ***		0.22	0.35
Metacognitive Beliefs	0.51 ***		0.44	0.57
Health Anxiety × Metacognitive Beliefs	0.01 ***		0.01	0.02
*R* ^2^		0.59		
*F*		234.45 ***		

*** *p* < 0.001. Abbreviations: CI, confidence interval; LL, lower limit; UL, upper limit.

**Table 7 jcm-11-02590-t007:** Moderation of metacognitive beliefs between health anxiety and excessiveness (cyberchondria construct) (*N* = 500).

Predictors	*Β*		Outcome: Excessiveness	
			95% CI	
			LL	UL
(constants)	21.57 ***		20.61	21.70
Health Anxiety	0.30 ***		0.24	0.37
Metacognitive Beliefs	0.48 ***		0.42	0.54
Health Anxiety × Metacognitive Beliefs	0.01 ***		0.01	0.02
*R* ^2^		0.58		
*F*		228.68 ***		

*** *p* < 0.001.

**Table 8 jcm-11-02590-t008:** Moderation of metacognitive beliefs between health anxiety and reassurance (cyberchondria construct) (*N* = 500).

Predictors	*Β*		Outcome: Reassurance	
			95% CI	
			LL	UL
(constants)	16.14 ***		15.75	16.53
Health Anxiety	0.20 ***		0.16	0.25
Metacognitive Beliefs	0.37 ***		0.33	0.42
Health Anxiety × Metacognitive Beliefs	0.01 ***		0.005	0.01
*R* ^2^		0.60		
*F*		249.87 ***		

*** *p* < 0.001. Abbreviations: CI, confidence interval; LL, lower limit; UL, upper limit.

## Data Availability

Related data are available from the first author upon reasonable request.

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
