# Peer review of "Relationship between Health-Anxiety and Cyberchondria: Role of Metacognitive Beliefs"

_jcm, 2022, doi:10.3390/jcm11092590_

Round 1

Reviewer 1 Report

Thank you for the opportunity to review the updated version of this manuscript. The authors have incorporated all feedback and I believe the paper is significantly improved as a result. However, I would still recommend some changes. I've made follow up notes a few of the previous feedback points, with the author responses in brackets:

2. I recommended thoroughly review this manuscript for grammatical errors, tense agreement, adherence to formal academic language, and redundancies or repetitive writing. [Corrected]

While the language is improved there are still numerous minor errors throughout the paper. They often involve improper verb tense. Some examples:

"a person have a heart attack in the past,” (Line 33)

“Such people research their symptoms online and they specifically focused on the information that suggest that they have a serious health-related problem.” (Line 60)

"This online health-related information seeking will not consider as cyberchondria until[...]" (Line 65)

4. Elaborate a bit on safety behaviors for those who aren’t familiar with the term. [Done see lines 50-53]

I’d reword this to clearly differentiate between the definition and the examples. Like: “Safety behaviors are efforts to mitigate anxiety or avoid a feared outcome. In the case of health anxiety, this can include reassurance-seeking behaviors such as repeatedly undergoing thorough medical exams and looking up health-related information.”

7. Suggest not using the term “final verdict”. [Correction done see line 80]

Recommend changing the revised term “final decision” to “definitive data” or “consensus on." 

13. Recommend providing clearer labels on the graphs or adding a key for the abbreviations. It could helpful to include a chart of the strength of the relationships in the moderation model. [Added]

Graphs are still inconsistently labeled and some use shorthand (“TotalCyb,” “Healthan”), rather than full labels or established abbreviations. See line 313, for example.

New feedback point: 

Lines 117-131 have been revised and are now very similar to the text in lines 644-657 of the discussion. Recommend cutting this text out of the discussion since it is not new information or discussion about the results.

Author Response

Cover Letter for 6th April 2022 review

Reviewer 1 Comments:

  1. While the language is improved there are still numerous minor errors throughout the paper. They often involve improper verb tenses.

Reply: Corrected

  1. Elaborate a bit on safety behaviours for those who aren’t familiar with the term. 

Reply: Done see lines 51-53

  1. Suggest not using the term “final verdict”. 

Reply: Correction done see line 80

  1. Recommend providing clearer labels on the graphs or adding a key for the abbreviations. It could help to include a chart of the strength of the relationships in the moderation model. Graphs are still inconsistently labelled and some use shorthand (“TotalCyb,” “Healthan”), rather than full labels or established abbreviations. See line 313, for example.

Reply: Correction done

  1. Lines 117-131 have been revised and are now very similar to the text in lines 644-657 of the discussion. Recommend cutting this text out of the discussion since it is not new information or discussion about the results.

Reply: Correction done. See lines 639-652

Reviewer 2 Report

The manuscript I received was not the final version (I assume), since it included revisions, tracked changes and highlights from the authors with some formatting notes.

Anyway, the text doesn't look as clear as it should be for a submitted manuscript. Basic English (grammatical and syntactic) errors are present throughout the text and even in the abstract, making it difficult to follow and assess. Please proof-read it before submitting.

Moreover, no ethical information is present about the study.

Author Response

Cover Letter for 6th April 2022 review

Reviewer 2 Comments:

  1. The text doesn't look as clear as it should be for a submitted manuscript. Basic English (grammatical and syntactic) errors are present throughout the text and even in the abstract, making it difficult to follow and assess. Please proofread it before submitting it.

Reply: Correction done

  1. No ethical information is present about the study.

Reply: Added see lines 210-213

Round 2

Reviewer 2 Report

Thank you for sending the revised version.

A couple of suggestions:

Page 2, line 87, please add in parenthesis "(CAS)" after the definition "cognitive attention syndrome" (and maybe use capital letters, like "Cognitive Attention Syndrome (CAS)" if you consider it useful), because some lines later (line 90) the "CAS" acronym appears out of nothing and readers might be wondering what does it mean.

I suggest merging the two sentences in [page 3, lines 141-143] in the same sentence: "Thus, it would be significant to understand the relationship between health anxiety, cyberchondria, and metacognitive beliefs, acknowledging it will help medical practitioners in treating cyberchondria".

Author Response

Dear Reviewer, thank You for Your important suggestions. We have corrected the paper as recommended (please see updated manuscript and yellow highlighting. All the best and take care, Domenico De Berardis and Irfan Ullah